# Genomic analysis and clinical correlations of non-small cell lung cancer brain metastasis

Anna Skakodub[1,2,11], Henry Walch [3,4,11], Kathryn R. Tringale[1,11], Jordan Eichholz[1], Brandon S. Imber [1], Harish N. Vasudevan[5,6], Bob T. Li [2,7], Nelson S. Moss[8], Kenny Kwok Hei Yu [8], Boris A. Mueller[1], Simon Powell [1], Pedram Razavi [2,7,9], Helena A. Yu [7,9], Jorge S. Reis-Filho [2,10], Daniel Gomez[1,2], Nikolaus Schultz [3,4,12] & Luke R. G. Pike [1,2,12] ✉

Up to 50% of patients with non-small cell lung cancer (NSCLC) develop brain metastasis (BM), yet the study of BM genomics has been limited by tissue access, incomplete clinical data, and a lack of comparison with paired extra-cranial specimens. Here we report a cohort of 233 patients with resected and sequenced (MSK-IMPACT) NSCLC BM and comprehensive clinical data. With matched samples (47 primary tumor, 42 extracranial metastatic), we show *CDKN2A/B* deletions and cell cycle pathway alterations to be enriched in the BM samples. Meaningful clinico-genomic correlations are noted, namely *EGFR* alterations in leptomeningeal disease (LMD) and *MYC* amplifications in multifocal regional brain progression. Patients who developed early LMD frequently have had uncommon, multiple, and persistently detectable *EGFR* driver mutations. The distinct mutational patterns identified in BM specimens compared to other tissue sites suggest specific biologic underpinnings of intracranial progression.

Lung cancer is a devastating disease that remains a leading cause of cancer-associated death and morbidity worldwide[1,2]. The standard treatment approach for limited BM is resection or stereotactic radiosurgery (SRS), although some targeted agents showed promising activity in the central nervous system (CNS). Patients with BMs, however, are often excluded from clinical trials of novel targeted agents given the unpredictable relationship between systemic and CNS responses.

The paucity of high-quality BM samples has limited efforts to understand the fundamental biology of BM, tropism, and biomarkers of CNS progression. Prior studies have sought to understand the

molecular characteristics of BM[3,4]. Whole exome sequencing (WES) of a heterogeneous cohort of 86 BMs, including tumors from breast, lung, and other primary histologic types[5] demonstrated branched evolution from the primary tumor to matched BMs while finding genetic homogeneity among spatially and temporally separated BMs. A more focused analysis of BM specimens from 73 NSCLC patients[6] revealed more frequent copy number alterations in *CDKN2A/B, MYC, YAP1*, and *MMP13* in BM specimens, as compared to a matched TCGA cohort. A recent larger-scale study evaluated 3035 NSCLC patients (67 of whom had paired BM and primary tumor samples) using a hybrid capture-based comprehensive genomic profiling assay[7]. They reported

[1]Department of Radiation Oncology, Memorial Sloan Kettering Cancer Center, New York, NY 10065, USA. [2]Biomarker Development Program, Memorial Sloan Kettering Cancer Center, New York, NY, USA. [3]Department of Epidemiology and Biostatistics, Memorial Sloan Kettering Cancer Center, New York, NY 10065, USA. [4]Marie-Josée and Henry R. Kravis Center for Molecular Oncology, Memorial Sloan Kettering Cancer Center, New York, NY 10065, USA. [5]Department of Radiation Oncology, University of California San Francisco, San Francisco, CA 94118, USA. [6]Department of Neurological Surgery, University of California, San Francisco, CA 94118, USA. [7]Department of Medicine, Memorial Sloan Kettering Cancer Center, New York, NY 10065, USA. [8]Department of Neurological Surgery, Memorial Sloan Kettering Cancer Center, New York, NY 10065, USA. [9]Department of Medicine, Weill Cornell Medical College, New York, NY 10065, USA. [10]Department of Pathology and Laboratory Medicine, Memorial Sloan Kettering Cancer Center, New York, NY, USA. [11]These authors contributed equally: Anna Skakodub, Henry Walch, Kathryn R. Tringale. [12]These authors jointly supervised this work: Nikolaus Schultz, Luke R.G. Pike. ✉e-mail: pikel@mskcc.org

alterations in various genes (such as *TP53*, *KRAS*, *CDKN2A* etc.) enriched in the BM cohort compared to unmatched primary sites. Unfortunately, sparse clinical outcomes were reported.

In the current analysis, we expanded on this prior work through molecular profiling and detailed clinical annotation on a large, homogenous cohort of NSCLC BM specimens with both matched primary tumor (PT) and extracranial metastasis (EM) samples. The main objectives were to (1) describe the unique molecular features of NSCLC BM and (2) identify genomic biomarkers associated with intracranial disease progression.

# Results

## Patient cohort
Of 233 patients, 133 (57%) were female, and the median age was 67 (Table 1; Supplementary Data File 1). The number of current and former smokers were 57 (25%) and 129 (55%), respectively. At the time of BM presentation, the median Karnofsky Performance Status (KPS) was 80 (range 40-100), and 212 (91%) had neurological symptoms, the most common of which were altered mental status,

**Table 1 | Patient and treatment characteristics**

| Patient characteristics | Total 233, *N* (%) |
|---|---|
| **Sex, No. (%)** | |
| Female | 133 (57) |
| Male | 100 (43) |
| **Smoking status, No. (%)** | |
| Current | 57 (25) |
| Former | 129 (55) |
| Never | 47 (20) |
| **Primary histology, No. (%)** | |
| Adenocarcinoma | 180 (77) |
| Squamous cell carcinoma | 23 (10) |
| Non-small cell, other | 30 (13) |
| Age, Median (range) | 67 (31–91) |
| KPS, Median (range) | 80 (40–100) |
| **Number of BM at resection, No. (%)** | |
| 1 | 117 (50) |
| 2–5 | 84 (36) |
| 6–15 | 30 (13) |
| >15 | 2 (1) |
| **Diameter of largest brain metastasis, cm** Median (range) | 3.0 (0.9–7.6) |
| **Neurologic symptoms at resection, No. (%)** | |
| Yes | 212 (91) |
| No | 21 (9) |
| **Treatment prior to resection** | |
| None, No. (%) | 122 (53) |
| **Systemic therapy[a], No. (%)** | 110 (47) |
| Cytotoxic chemotherapy | 71 (65) |
| Immunotherapy | 20 (18) |
| Tyrosine Kinase Inhibitor | 13 (12) |
| VEGF Inhibitor | 3 (3) |
| Other | 3 (3) |
| **Radiation Therapy, No. (%)** | 16 (7) |
| Stereotactic radiosurgery | 11 (69) |
| Whole-brain radiotherapy | 4 (25) |
| Prophylactic cranial irradiation | 1 (6) |

[a]Received either monotherapy or combination therapy as the most recent therapy prior to resection.

ataxia, and motor weakness. Many (122, 52%) patients were treatment-naive prior to BM resection; 110 (47%) received systemic therapy prior to craniotomy (median number of systemic therapy lines, 1 [range 1–8]). Few (16, 7%) patients had brain-directed radiotherapy before BM resection.

## Comparison of genomic differences between BM and non-BM specimens
The TMB was significantly higher in the BM specimens compared to other extracranial metastases (BM median: 8.8, extracranial median: 5.8; $p = 0.00766$; Fig. 1B). The FGA was also significantly higher in the BM samples compared to either extracranial metastases or the primary site tissue sample (BM vs. extracranial metastases: $p = 2.765e{-}06$; BM vs. primary: $p = 2.273e{-}07$; Fig. 1B).

When comparing mutations, copy-number alterations (CNAs, i.e., amplifications and deletions), and structural variants (i.e., rearrangement and fusions) between the BM, EM, and PT specimens, *CDKN2A/B* alterations were more common in the BM samples (34%) compared to PT (13% $p = 0.003$, $q = 0.04$; Fig. 1C; Supplementary Data File 2). A similar representation of alterations was identified in other cancer-related genes (e.g., *TP53*, *KRAS*, and *EGFR*) in the BM specimens as in the EM and PT. *MYC* alterations were not enriched in the BM specimens compared to the other two groups.

At the pathway-level, cell cycle pathway alterations were more common in the BM specimens compared to the PT specimens (56% vs. 32%, $p = 0.004$, $q = 0.041$; Fig. 1D). This effect was driven by differences in *CDKN2A/B* alterations[8]. When genome-wide CNAs were examined among the three groups, a higher amount of chromosomal instability was observed in the BM samples compared to the other groups (Fig. 1E).

## Stratified analyses by histologic subtype
When we compared gene and pathway alterations seen in the BM specimens, stratified by histology (LUAD, squamous cell carcinoma [SCC], and other NSCLC) we noted more frequent *KRAS* and *STK11* alterations (*KRAS*: 35% vs 9%, $p = 0.009$, $q = 0.049$; *STK11*: 22% vs 0%, $p = 0.01$, $q = 0.049$), as well as RTK-Ras pathway alterations in LUAD BM samples as compared to the SCC BM samples (86% vs 57%, $p = 0.002$, $q = 0.022$) (Suppl. Fig. 1A). *CDKN2A* deletions were more frequent in SCC group as compared to LUAD group. Examination of genome-wide CNAs across histologies revealed markedly varying CNA profiles (Suppl. Fig. 1B, C), consistent with previously reported results[9].

Thus, to mitigate potential confounding from primary tumor histology, further analyses were performed exclusively in the LUAD cohort (180 of 233, 77%). One other sample was excluded from further genomic analyses due to a high degree of microsatellite instability (MSI). Therefore, 179 BM, 37 PT, and 34 EM samples were included in subsequent analyses. The overall makeup of this sub-cohort was like that of the entire cohort (Suppl. Table 1). Similarly, FGA was significantly higher in LUAD BM compared to EM or PT (Supp. Fig. 1C). Analogous to the total NSCLC cohort, *CDKN2A/B* alterations and cell cycle pathway alterations remained enriched in the BM LUAD group compared to PT and EM (*CDKN2A/B*: 31% vs 18%, $p = 0.004$, $q = 0.14$; cell cycle pathway: 52% vs 27%, $p = 0.007$, $q = 0.072$) (Suppl. Fig. 1D; Supplementary Data File 3; Supplementary Data File 4).

## Genomic biomarkers of CNS tropism
To assess associations between PT genomic profiles and development of BM or EM, three distinct cohorts of LUAD PT samples were compared as outlined above: (1) PT LUAD BM+ ($N = 32$), (2) PT LUAD BM−, EM+ ($N = 1549$), and (3) PT LUAD BM−, EM− ($N = 582$)[10]. Alterations in *TP53*, *MYC*, *SMARCA4*, *RB1*, *ARID1A*, and *FOXA1* were significantly enriched in PT specimens from patients who developed BM compared to those who did not have BM (Suppl. Fig. 1E). *NKX2-1* alterations were also enhanced in both BM and EM cohorts compared to patients

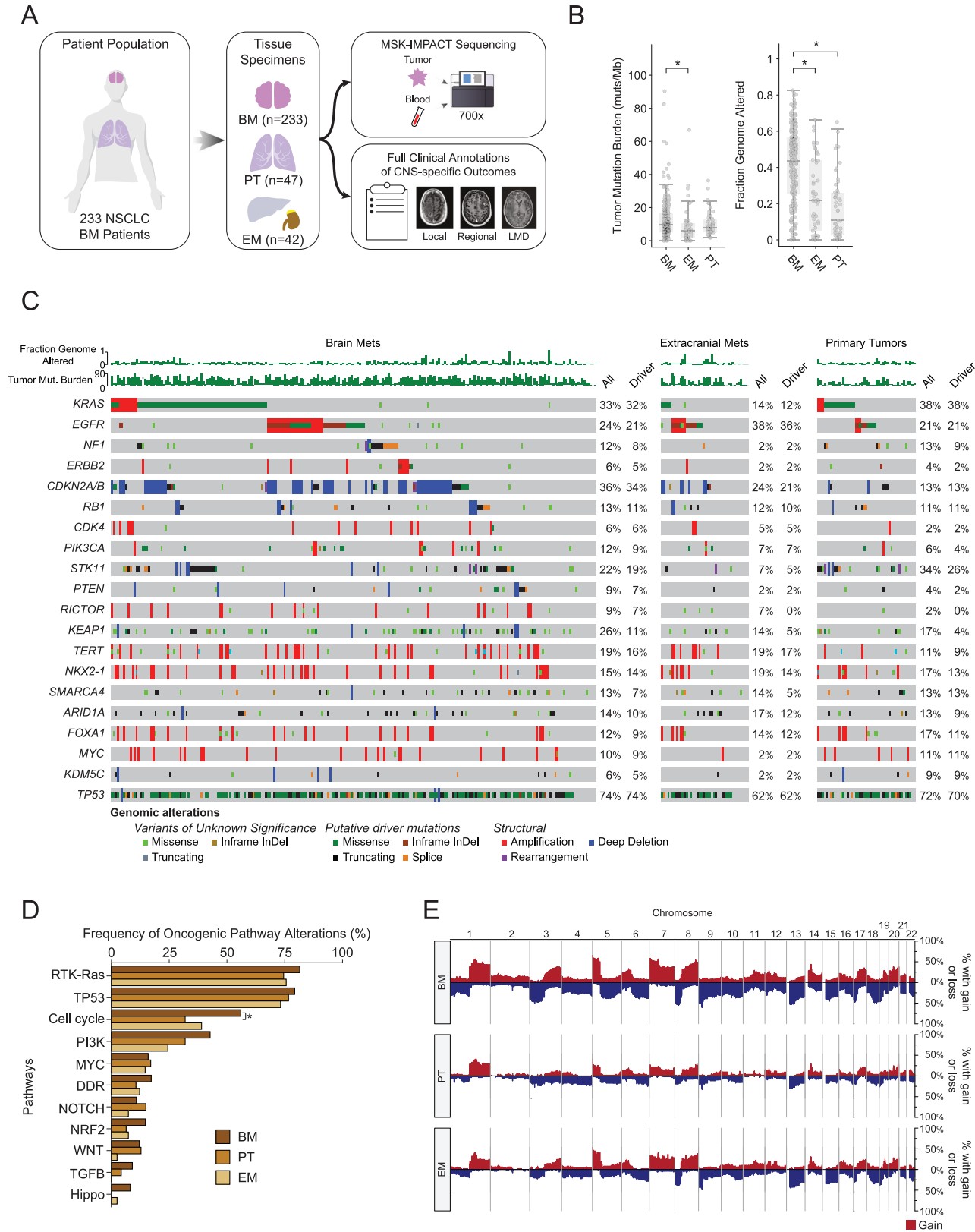

without metastatic disease. In addition, we found MYC pathway alterations were enriched in patients with BM development compared to patients without metastatic disease, and TP53 and DNA damage repair pathway alterations were significantly enriched in those with BM and EM compared to patients without metastatic disease (Suppl. Fig. 1E).

### Genomic correlates of paired analysis

We next performed detailed pairwise comparisons of matched specimens, collected asynchronously or synchronously as described above. Interestingly, patients who had BM resection followed by EM or PT biopsy, and patients who had an initial tissue collected from EM/PT, and subsequently developed BM demonstrated many alterations

**Fig. 1 | Study design and genomic differences between BM NSCLC and primary tissue (PT) or extracranial metastatic (EM) sites. A** Overview of study design. **B** Comparison of broad genomic features between brain metastases (BM) samples ($n = 233$), extracranial metastases (EM) samples ($n = 42$), and primary tumor (PT) samples ($n = 47$) (TMB comparison: BM vs. extracranial median: 5.8; $p = 0.00766$; FGA comparison: BM vs. extracranial metastases: $p = 2.765e\text{-}06$, BM vs. primary: $p = 2.273e{-}07$). A two-sided Mann–Whitney $U$-test was used to assess statistical significance. The center line of the box plots indicates the median. The bounds of the box indicate the interquartile range. The whiskers indicate the highest and lowest values not considered outliers. Asterisks indicate significance between groups being compared. **C** Oncoprint depicting the most frequent oncogenic alterations in BM, EM, and PT samples. **D** Comparison of oncogenic signaling pathway alterations across BM, EM, and PT samples. The cell cycle pathway was significantly enriched in BM vs PT tumors ($p = 0.004$, $q = 0.041$). A two-sided Fisher's exact test was used to assess statistical significance. Multiple hypotheses testing was performed using a Benjamini-Hochberg correction. Asterisks indicate significance between groups being compared. **E** Genome-wide copy number profiles for BM, PT, and EM samples. Source data are provided as a Source Data file for Fig. 1.

unique to the BM specimens (Fig. 2A, B). *TP53* (34%) *and EGFR* (27%), alterations were commonly identified alterations shared between BM and later PT/EM samples (Fig. 2A; Suppl. Fig. 2A). In contrast, alterations in *TP53* and *KRAS* were often present at diagnosis and retained in the PT/EM and BM specimens of patients who developed BM later in their clinical course (Suppl. Fig. 2B). We likewise identified a subset of patients whose BM specimens had acquired private mutations in *HLA-B* (Fig. 2B).

When we compared matched pairs of BM and subsequently acquired cerebrospinal fluid (CSF) specimens, we noted that some BM specimens had unique alterations in *TP53* and *KRAS*, but there were notably very few unique mutations in the CSF specimens (Fig. 2C; Suppl. Fig. 2C). Among patients with simultaneous collection of BM and PT, most alterations were unique to BM or PT (Fig. 2D); however, this finding is limited by sample size ($N = 2$). We were able to identify a subset of nine patients in whom we had multiple BM specimens. Seven of these patients had two independent lesions resected. Interestingly and in contrast to the synchronous BM/PT specimens, we found high concordance in the genomic profiles in these BM-BM pairs (Fig. 2E).

Finally, we identified two patients with three specimens collected through their illness. Remarkably, in one patient who had a PT followed by a BM and then a separate PT sequenced, we identified numerous driver mutations, none of which were shared; by contrast, in another patient who had an EM, then BM, and then a PT biopsied, we noted shared driver mutations in *EGFR* and *TP53* (Fig. 2F). In this patient, there was evidence of acquired resistance in the BM specimen, identifying an *EGFR* T790M mutation in the BM specimen that was retained in the subsequent PT specimen.

## Genomic correlates with clinical presentation and prior therapy

We next sought to compare the genomic profiles of BM from patients who: presented with BM as a progression event vs. at diagnosis; had multiple lesions vs. a single lesion; who had received prior chemotherapy vs. those that did not; and lastly, those that received TKI vs. those that did not. As expected, *EGFR* alterations were more common and *KRAS* mutations were less common among patients who received prior TKI treatment, but we did not identify any other statistically significant differences in driver mutations between groups (Fig. 3A).

## Genomic biomarkers of intracranial disease progression

Most (101, 56%) LUAD patients with BM experienced intracranial POD following initial craniotomy and RT, most frequently as regional progression (54, 30%), followed by local progression (25, 14%), and LMD (20, 11%). Two patients had unclear intracranial disease progression patterns and were excluded from the cohort. The median OS and iPFS from BM diagnosis was 2.7 years (95%CI 2.3–4.0) and 1.2 years (95%CI 1.0–1.5), respectively (Fig. 3B, C).

To evaluate genomic biomarkers of intracranial disease progression, we grouped patients by pattern of progression and looked for differences in driver mutation frequency (Fig. 3D). We found that patients in the LMD cohort were more likely to have *EGFR* alterations as compared to the non-progressor group (45% vs 21%, $p = 0.044$, $q = 0.789$). By contrast, patients with local progression had more

frequent *RB1* loss (24% vs. 6%, $p = 0.022$, $q = 0.573$) or *NKX3-1* alterations (16% vs. 3%, $p = 0.044$, $q = 0.573$) as compared to the non-progressor group. Likewise, *MYC* amplifications were more common in patients who later suffered multifocal regional progression, compared to those with local progression, where no *MYC* amplifications were detected (22% vs 0%, $p = 0.023$, $q = 0.790$). There was no statistically significant difference in *CDKN2A/B* alterations across the five cohorts (Fig. 3D). *NKX2-1* had a higher amplification frequency (22%) in patients without intracranial disease progression than those with local progression or LMD (4% and 10 %, respectively). We also noted more frequent alterations in *NF1* in patients who developed LMD (15%) as compared to other groups (Suppl. Fig. 2F).

Upon assessing frequencies of oncogenic pathway alterations, MYC pathway alterations were significantly enriched in the patients with LMD ($p = 0.013$, $q = 0.14$) and regional progression (both single: $p = 0.023$, $q = 0.255$, and multifocal: $p = 0.023$, $q = 0.255$) compared to local progression (Fig. 2E). Most cases appear to require cell cycle pathway alterations for initial BM progression (Suppl. Fig. 2D). However, these alterations do not influence patterns of POD (Suppl. Fig. 2E). Alteration frequencies within the RTK and RAS pathways were assessed across progression patterns to identify concurrent events. *EGFR* and *KRAS* were the most frequently altered genes (Suppl. Fig. 2F). Assessment of WGD events across the progression groups revealed that patients with LMD had the numerically highest WGD frequency (Suppl. Fig. 2G).

## EGFR alterations in patients with LMD

Given the clear enrichment in *EGFR* alterations in patients with LMD, this finding was further investigated. Patients who suffered from LMD frequently exhibited less common *EGFR* mutations (45%), such as L861Q, G719A/S, A755G, or N771_H773dup (Fig. 4A).

We next identified patients with LMD as an initial form of disease progression who had multiple tissue samples collected throughout their disease course for more in-depth evaluation. We identified that above-described uncommon *EGFR* mutations were persistent in various tissue samples despite various therapies. For example, the first patient presented with BM at the time of initial lung cancer diagnosis and underwent craniotomy (Fig. 4B). This BM specimen contained *EGFR* L861Q and G719S driver mutations. After BM resection and postoperative RT, the patient received erlotinib, but developed systemic progression, with repeat lung biopsy revealing a known gatekeeper mutation (*EGFR* T790M); *EGFR* L861Q and G719S remained persistent. Systemic therapy was switched to osimertinib, and eventually, the patient had further systemic progression with contemporaneous LMD; additional biopsy specimens demonstrated clearance of the T790M mutation but ongoing presence of the L861Q and G719S mutations.

In another example (Fig. 4C), a patient presented with BM at initial lung cancer diagnosis and underwent BM resection. The BM specimen contained an *EGFR* exon-19 deletion (E746_A750del). The patient received postoperative RT followed by osimertinib and chemotherapy but still developed early LMD. CSF sampling showed elevated circulating tumor cells (CTCs) that were cleared after proton craniospinal irradiation, but multiple serial CSF samples showed persistence of the

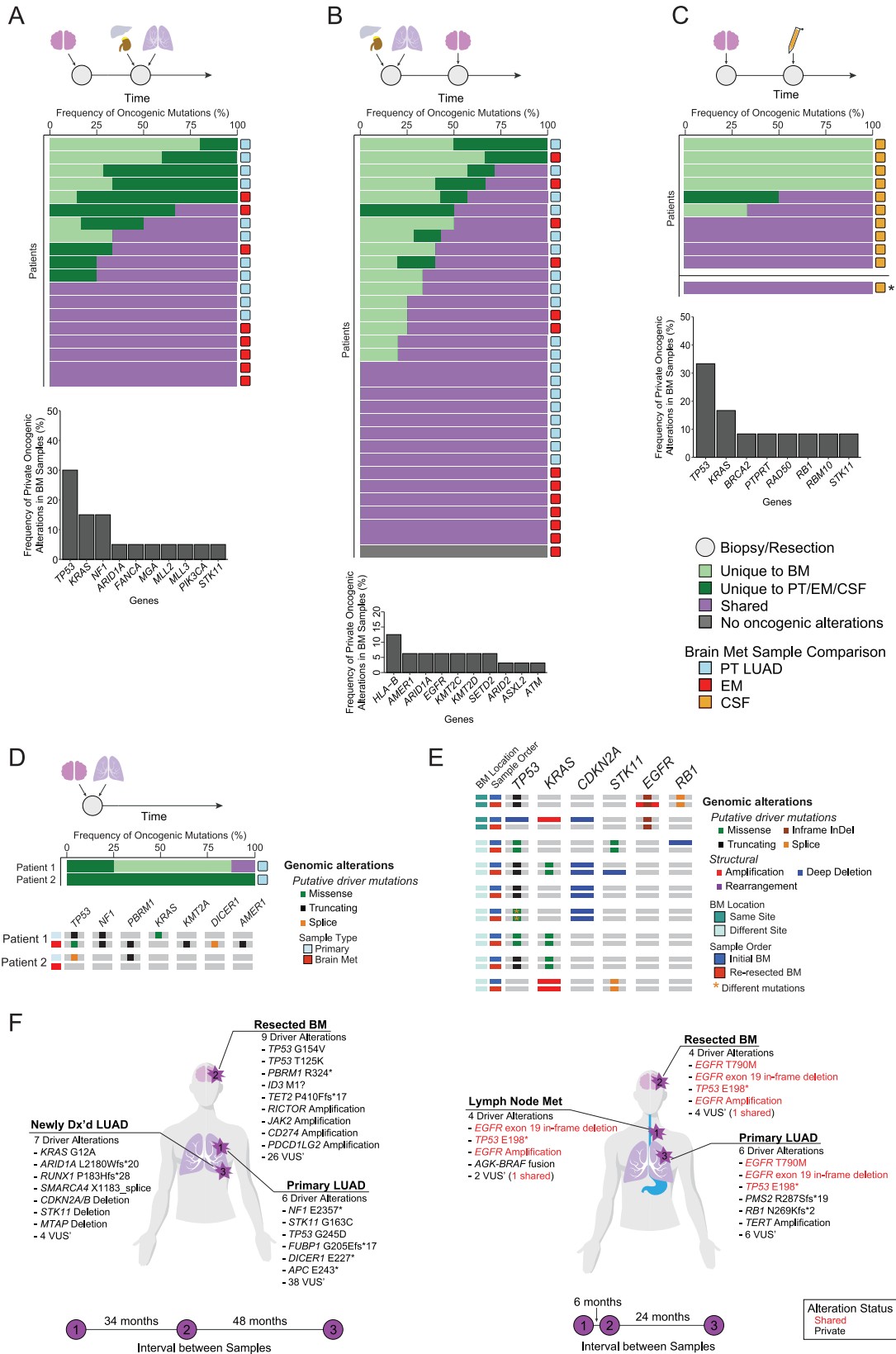

*EGFR* exon-19 deletion and a *TP53* R273L mutation until the patient succumbed to neurologic disease.

## Discussion

In this work, we present a detailed analysis of the genomic features and clinical correlates of a large cohort of NSCLC BM patients with matched extracranial and serially collected samples. We demonstrate that NSCLC BM are markedly altered compared to extracranial metastatic or primary disease, with higher TMB, FGA, and WGD seen in BM specimens. We confirm prior reports indicating cell cycle alterations, such as deep deletions in *CDKN2A/B*, are a common molecular feature of BM. Through the comparison of matched pairs

**Fig. 2 | Paired analysis. A** Overview of mutations that were either shared or unique when comparing BM to PT/EM samples when BM samples were obtained before PT/EM samples; the bar plot at the bottom represents the most frequently mutated genes that were private to the BM samples. **B** Overview of mutations that were either shared or unique when comparing BM to PT/EM samples when BM samples were obtained after PT/EM samples; the bar plot at the bottom represents the most frequently mutated genes that were private to the BM samples. **C** Overview of mutations that were either shared or unique when comparing BM to CSF samples when BM samples were obtained before CSF samples; the asterisk indicates one patient in which CSF was obtained before BM sample. The bar plot at the bottom

represents the most frequently mutated genes that were private to the BM samples. **D** Shared and unique mutations between patients with synchronous BM and PT/EM tumors. Oncoprint depicts the types of mutations across the samples per patient. **E** Oncoprint of BM tumor pairs from patients with multiple BM samples showing shared and unique alterations. **F** Patient vignettes for two patients with multiple samples per patient. Tumor locations are shown in the body maps and the intervals of time between samplings are depicted at the bottom. Oncogenic alterations identified for each tumor are written out, colored by whether they were shared or unique. Source data are provided as a Source Data file for Fig. 2.

of BM-EM/PT specimens, we noted generally high genomic concordance although uncommon private alterations of potential significance were noted in BM specimens. In an integrated analysis, we correlated brain-specific clinical outcomes with genomic alterations; provocatively, we found that patients who suffered leptomeningeal disease were more likely to have BM specimens with non-canonical *EGFR* mutations, which were persistent despite maximal EGFR-directed and local therapy.

This work is part of ongoing efforts to understand the biological underpinnings of BM across various cancer types. Common events that appear important for CNS progression include chromosomal instability, impaired DNA repair, copy number alterations, and cell cycle alterations. Specifically, copy number deletion of *CDKN2A* has been one of the most frequently reported events[11]. CDKN2A can inactivate the RB protein by binding to and inactivating the cyclin D-cyclin-dependent kinase four (cdk4) complex. The expression of this gene can cause cell cycle arrest in the G1 phase, inhibit cell proliferation, promote tumor cell apoptosis, and increase tumor cell chemotherapy sensitivity. The current study confirms frequent loss of *CDKN2A/B* and concordant cell cycle pathway alterations in NSCLC BM. Furthermore, ~50% of patients from this cohort had CNA in cell cycle genes that were non-overlapping and mutually exclusive, suggesting that this is an essential event in the development of brain metastasis. In this study, BM specimens showed global changes, including increased CNA, FGA, and TMB compared to extracranial specimens, which is in agreement with prior reports[12,13] suggesting divergent and branched evolution of BMs[5]. By contrast, we noted concordance of alterations in oncogenes and tumor suppressors such as *TP53*, *KRAS*, or *EGFR*, suggesting that these are essential, and independent of tumor microenvironment (TME). BM-specific cell cycle alterations may offer opportunities for targeted therapies such as CDK4/6 inhibitors[14], which is the focus of ongoing trial work.

We performed a detailed analysis of patient-matched BM-PT/EM pairs. Most mutations were present in both BM and matched PT/EM samples. Although underpowered to explore fully, we noted instances of BM private mutations with potential functional relevance; for example, several patients who developed BM as a form of treatment failure had acquired driver alterations in *HLA-B*. Homozygous deletions in *HLA-B* have previously been reported to confer acquired resistance to immune checkpoint inhibitors (ICIs) in LUAD[15], and other work has suggested *HLA-B* downregulation as a means by which metastatic clones escape T-lymphocyte and NK cell-mediated cytotoxicity[16]. In the context of recent single-cell sequencing data showing that the brain TME is characterized by reduced antigen presentation and B/T-cell function and increased M2-type macrophage activity[17], *HLA-B* alterations in LUAD cells may be permissive for cancer cell growth in the brain TME[18].

This study offers unique integration of CNS-specific clinical outcomes with genomic alterations in a large cohort of patients. We identified specific alterations that correlated with patterns of failure: we found *MYC* amplification to be associated with multifocal regional failure, whereas *RB1* deletions and *NKX3-1* alterations were associated with local disease progression. Mouse models of brain metastasis have indicated that overexpression of *MYC* promotes tumor cell

dissemination in brain tissues through protection against oxidative stress[19]. The association of *RB1* with local failure is puzzling since one might expect *RB1* loss to sensitize residual microscopic disease to adjuvant radiation therapy[20]; however, co-occurrence of *RB1* loss with other mutations might promote RT resistance. *NKX3-1* is less understood within the context of NSCLC but is associated with metastatic disease in prostate cancer[21]. With further validation, such findings could represent potential predictive biomarkers and inform therapeutic selection.

Finally, patients who suffered LMD as a first form of intracranial failure were far more likely to have *EGFR* alterations in BM specimens. Many of these alterations were uncommon drivers and continued to be detectable in serial samples despite maximal therapy with *EGFR*-directed TKIs and RT. Prior work has demonstrated that patients with atypical *EGFR* alterations receive lesser benefit from Osimertinib, with shorter overall survival[22]. More generally, it is known that EGFR-mutant NSCLC patients are predisposed to LMD[23]. The metabolic and microenvironmental features of CSF are markedly different from brain parenchyma[24]; thus, activating *EGFR* mutations may offer a means of spreading to and surviving in this otherwise nutrient-poor environment. Thus, the result that patients with non-canonical *EGFR* mutations in their resected BM specimens were more likely to fail with LMD rather than other forms of intracranial failure may be reflective of the combined effects of partial therapeutic resistance to Osimertinib and inherent tropism for the leptomeninges, in the context of a cohort of patients with otherwise excellent brain control and longer overall survival than non-oncogene driven NSCLC[25].

This study is limited by its retrospective design of a highly selected group of NSCLC patients with limited BM that were large and symptomatic, who therefore required surgical resection; thus, the genomic profiles and clinical outcomes for such patients may differ significantly from those with more extensive disease at diagnosis. Molecular data were obtained from routine clinical NGS (MSK-IMPACT), and thus only known cancer-associated genes were interrogated. Future work will include whole-exome DNA and whole-transcriptome RNA sequencing to identify potentially relevant non-coding elements, lesser-known somatic alterations, and transcriptional programs that are critical for the development and progression of brain metastasis.

## Methods
### Patient population
The cohort consisted of 233 patients with a history of NSCLC BM who underwent therapeutic craniotomy at a single center from January 2010 until April 2021 (Fig. 1A). The use of specimens for this study was approved by the institutional review board at MSK (protocols 06-107, 12-245, 16-314, and 23-051). All patients provided written informed consent for tumor sequencing and review of patient medical records for detailed demographic, pathologic, and treatment information. Complete clinical information was collected for all patients, including baseline characteristics, prior systemic therapy, radiotherapy (RT), and intracranial-specific clinical outcomes. In addition to the NSCLC BM samples, 47 PT samples and 42 EM samples from the same patients

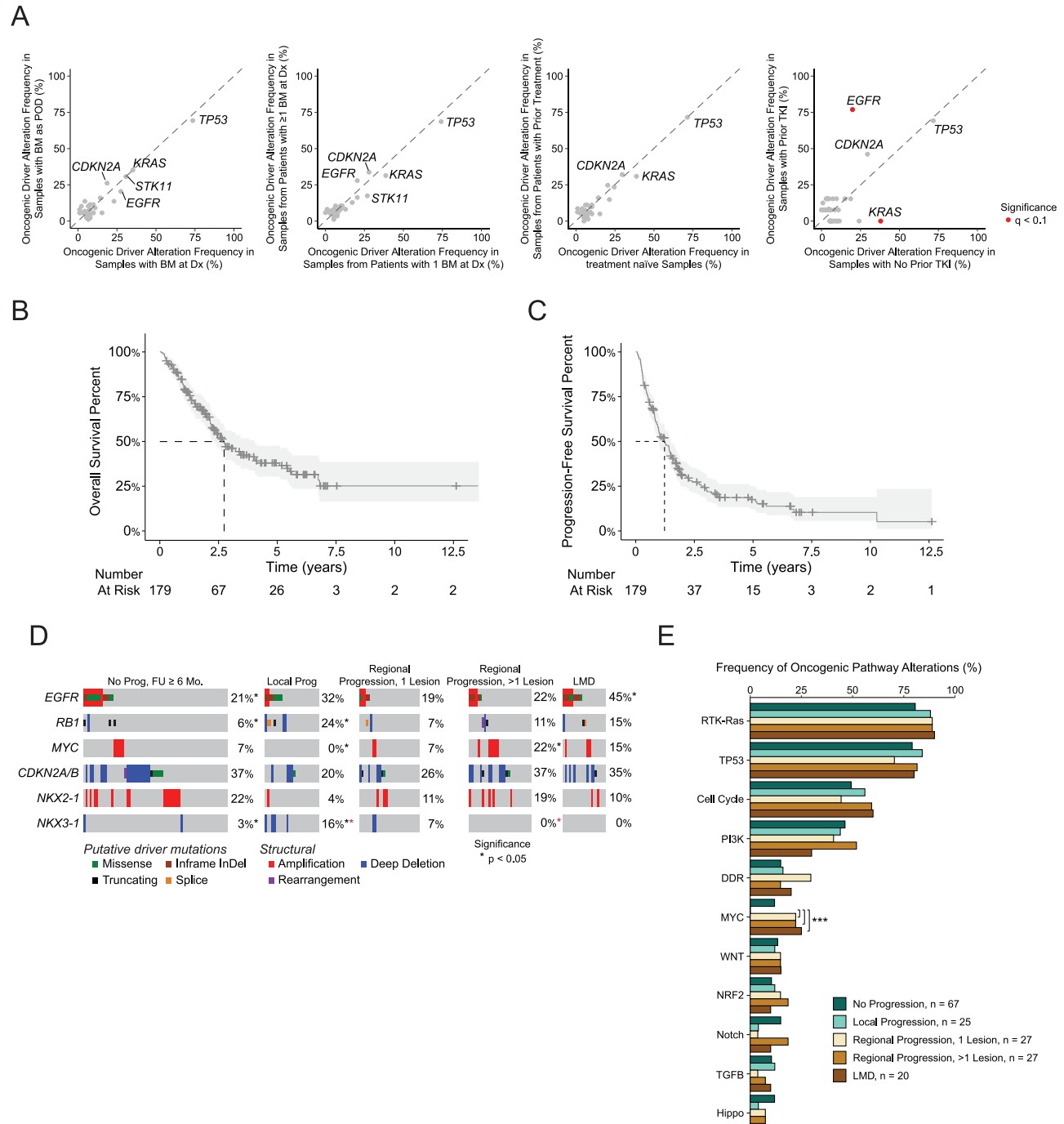

**Fig. 3 | Clinical and genomic correlates including disease progression in BM LUAD cohort. A** Scatterplots comparing driver alteration frequencies between (left to right): BM samples found at diagnosis versus BM samples found as progression of disease, BM samples from patients with one BM at diagnosis versus BM samples from patients with multiple BMs at diagnosis, treatment naïve BM samples versus BM samples from patients with prior treatment, and BM samples from patients with no prior tyrosine kinase inhibitor (TKI) treatment versus BM samples from patients with prior TKI treatment. Genes altered in at least 25% of one of the groups being compared are shown and red coloring of a point indicates significance. **B** Overall survival (OS) in BM LUAD group from the time of BM diagnosis. **C** Progression-free survival (PFS) in BM LUAD group from the time of BM diagnosis. **D** Comparison of oncogenic alterations in BM samples from patients with different types of intracranial disease progression. Comparisons with significant p-value results are shown with the presence of an asterisk by their alteration frequency. The color of the asterisk indicates which groups were being compared. **E** Pathway-level alterations between BM samples from patients with different types of intracranial disease progression. The MYC pathway was significantly enriched in the patients with LMD ($p = 0.013$, $q = 0.14$) and regional progression (both single: $p = 0.023$, $q = 0.255$, and multifocal: $p = 0.023$, $q = 0.255$) compared to patients with local progression. A two-sided Fisher's exact test was used to assess statistical significance. Asterisks indicate significance between groups being compared. Source data are provided as a Source Data file for Fig. 3.

were analyzed. EM samples included extracranial metastatic tissue and/or CSF samples. Sub-cohort analyses were performed on patients with lung adenocarcinoma patients (LUAD) only to remove histology as a potential confounding variable.

## Paired samples analyses

To evaluate the temporal relationship between metastases, paired samples with BMs were grouped by the timing of collection: (1) Synchronous specimens with contemporaneous collection of both BM

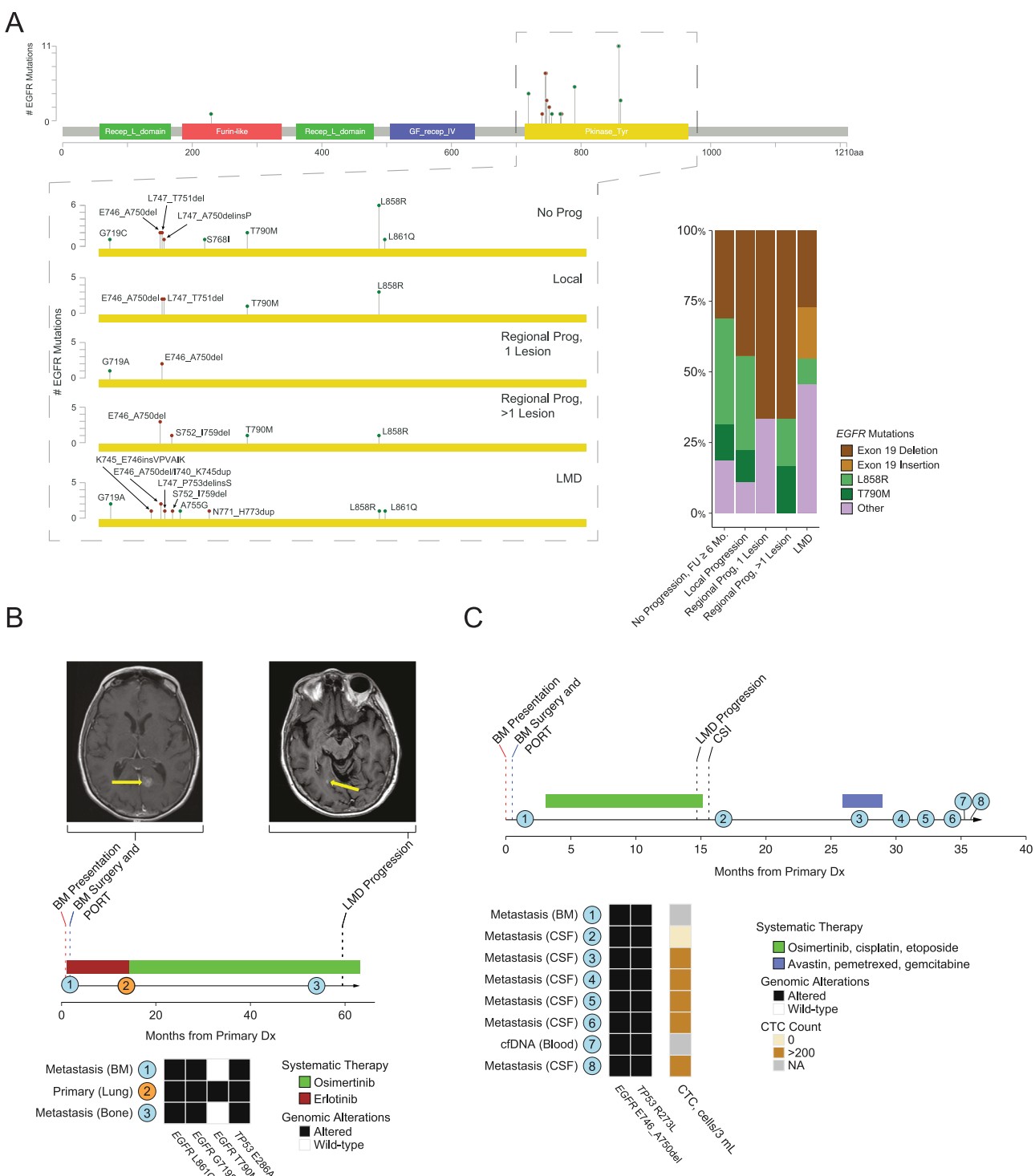

**Fig. 4 | *EGFR* alteration distributions and individual patient cases. A** Lollipop plot (on the left) of EGFR depicting the most common sites of mutations in the BM samples. The kinase domain is blown out to show the types of mutations by the type of intracranial progression. The stacked bar plot (on the right) depicts the most common types of mutations stratified by the type of intracranial progression. **B** Vignette of patient B with three sequenced samples. The disease timeline depicting the treatment the patient received and tumor samplings is shown beneath, along with what oncogenic alterations were shared or unique to each of the samples. **C** Vignette of patient C with multiple sequenced samples. The disease depicting the treatment the patient received and tumor samplings is shown beneath, along with what oncogenic alterations were shared or unique to each of the samples and the circulating tumor cells (CTC) count at each sampling. Source data are provided as a Source Data file for Fig. 4.

and EM/PT (within 60 days), (2) Intracranial progressors who had initial EM or PT collection followed by a craniotomy (>60 days later), and (3) Intracranial presenters who had a therapeutic craniotomy at diagnosis followed by systemic progression and re-biopsy of an EM or PT specimen (>60 days after craniotomy). We also identified patients who had both BM and CSF collected, and those who had multiple BM specimens (either multiple independent specimens or locally recurrent disease).

## Brain-specific clinical outcomes

Brain-specific clinical outcomes were defined based on standard approaches to clinical practice. Five distinct intracranial disease progression outcomes included: (1) no evidence of intracranial progression (POD) for at least 6 months of clinical follow-up, (2) local progression (i.e., clear evidence of regrowth of the initially resected lesion with orthogonal imaging in the form of PET brain or perfusion to confirm active disease, as opposed to radionecrosis/treatment effect), (3) regional progression with a single new lesion (i.e., a new solitary lesion outside of the resected intracranial cavity), (4) multifocal regional progression (i.e., more than one lesion outside of the resected intracranial cavity) and (5) leptomeningeal disease (LMD) development (clear evidence confirmed by contrast-enhanced MRI brain with corroborating neurologic symptoms and/or positive CSF cytology). In cases of mixed POD patterns, patients with regional and local progression were considered regional POD, and patients with LMD with simultaneous concern for local or regional POD were considered to have LMD. Radiographic POD was called per the above clinical criteria by a board-certified neuroradiologist, often reviewed at a multidisciplinary tumor board, with the use of orthogonal imaging (contrast-enhanced MRI brain combined with perfusion, PET, delayed contrast, or spectroscopy) and pathologic data, and verified by documentation of a change in clinical management in subsequent medical or radiation oncology notes.

To assess whether underlying genomic profiles of PT in patients with LUAD are associated with BM development, LUAD PT samples from the paired analysis were compared to two distinct institutional LUAD PT cohorts[10]. In this manner, three distinct cohorts of LUAD PT samples were formed: (1) patients who developed metastatic disease with intracranial involvement (PT LUAD BM+), (2) patients who developed only extracranial metastatic disease (PT LUAD BM−, EM+), and (3) patients who never developed metastatic disease of any sort (PT LUAD BM−, EM−).

## Genomic analysis

All samples were evaluated using Memorial Sloan Kettering-Integrated Molecular Profiling of Actionable Cancer Targets (MSK-IMPACT) assay[26]. This is a custom FDA-authorized next-generation sequencing (NGS)-based assay that uses a paired-sample analysis pipeline to identify somatic variants in the targeted exons with an average coverage depth of 700x. Tumor DNA was sequenced using one of four versions of MSK-IMPACT (IMPACT 341, IMPACT 410, IMPACT 468, or IMPACT 505). A matched normal sample (blood) was used in all cases. Genomic alterations were filtered for oncogenic events using OncoKB[10] Genes were consolidated into pathways using curated templates from the TCGA[27]. Germline alterations were excluded from this analysis. Tumor mutational burden (TMB) was defined as the number of nonsynonymous mutations per megabase covered by the IMPACT panel. The fraction genome altered (FGA) was defined as the length of the sequenced genome with a log2 copy number variation (gain or loss) >0.2 divided by the total size of the genome profiled for copy number. The FACETS (Fraction and Allele-Specific Copy Number Estimates from Tumor Sequencing) algorithm[28] and the FACETS-suite package (https://github.com/mskcc/facets-suite) were used to generate purity-corrected fraction of genome altered estimates and assess whole-genome duplication (WGD). Tumors were considered to have undergone WGD if at least 50% of their autosomal genome had a major copy number of 2 or more[29].

## Statistical analysis

Baseline clinical characteristics and genomic alteration frequencies were compared using a two-sided Fisher's exact test. Continuous variables were compared using a Wilcoxon test. Kaplan–Meier curves were generated using overall survival (OS) and intracranial progression-free survival data (iPFS). Multiple testing correction was performed using the Benjamini-Hochberg method (*q*-value cutoff of 0.1). All analyses were performed using R v3.6.1.

## Reporting summary

Further information on research design is available in the Nature Portfolio Reporting Summary linked to this article.

## Data availability

The raw sequencing data for the MSK-IMPACT analysis is protected and cannot be broadly available due to privacy laws; patient consent to deposit raw sequencing data was not obtained. De-identified data are available under restricted access to protect patient privacy in accordance with federal and state law. Raw data may be requested from schultzn@mskcc.org with appropriate institutional approvals. Data will be shared for a span of 2 years within 2 weeks of execution of a data transfer agreement with MSK, which will retain all title and rights to the data and results from their use. All de-identified clinical and genomic data for the patients in this study have been deposited in the cBioPortal for Cancer Genomics[9,30] and are publicly available for browsing and download at https://www.cbioportal.org/study/summary?id=bm_nsclc_mskcc_2023. All other data generated in this study are available within the article and its supplementary data files. Source data are provided with this paper.

## Code availability

The FACETS-suite R package (https://github.com/mskcc/facets-suite) and the OncoKB annotator tool (https://github.com/oncokb) are available on GitHub. The MSK-IMPACT data analysis pipeline, as well as additional custom programs and tools are available on the MSK GitHub repository at https://github.com/mskcc.

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

## Acknowledgements

The authors would like to express their gratitude to all individuals that have contributed to this research. We sincerely thank the participants who took part in this study, as their involvement was essential to its success. We also extend our appreciation to our colleagues for their valuable insights, discussions, and technical support throughout the project.

## Author contributions

Conceptualization A.S., H.W., K.R.T., L.R.P., and N.S.; Methodology A.S., H.W., K.R.T., and L.R.P.; Validation K.R.T., D.G., S.P., L.R.P., and N.S.; Formal analysis A.S., H.W., and K.R.T.; Data curation A.S., H.W., K.R.T., and J.E.; Visualization H.W.; Writing—Original draft, A.S., H.W., K.R.T., and L.R.P.; Writing—Review and editing A.S., H.W., K.R.T., B.S.I., H.N.V., B.T.L., N.S.M., K.K.H.Y., B.A.M., S.P., Razavi, H.A.Y., J.S.R., D.G., N.S., and L.R.P.; Final approval of manuscript: A.S., H.W., K.R.T., J.E., B.S.I., H.N.V., B.T.L., N.S.M., K.K.H.Y., B.A.M., S.P., Razavi, H.A.Y., J.S.R., D.G., N.S., and L.R.P.; Supervision, L.R.P. and N.S.

## Competing interests

B.S.I.: GT Medical Technologies, Inc., Provision of Services; B.T.L.: Amgen, Provision of Services (uncompensated), Asia Society, Provision of Services (uncompensated), AstraZeneca, Provision of Services (uncompensated), BeiGene, Ltd., Provision of Services (uncompensated), Bolt Biotherapeutics, Inc., Provision of Services (uncompensated), Daiichi Sankyo, Provision of Services (uncompensated), Karger Publishers Intellectual, Property Rights, Roche, Provision of Services (uncompensated), Shanghai Jiao Tong University Press Co., Ltd., Intellectual Property Rights; N.S.M.: AstraZeneca, Provision of Services; K.K.H.U.: Aptorum Group Limited, Ownership / Equity Interests; S.P.: PharmaPier US LLC, Provision of Services (uncompensated), Rain Therapeutics Inc., Provision of Services, Varian Medical Systems, Provision of Services; Razavi: Biovica, Provision of Services, Inivata, Inc., Provision of Services, Novartis, Provision of Services, Tempus Labs, Inc., Provision of Services (uncompensated); H.A.Y.: AstraZeneca, Provision of Services, Black Diamond Therapeutics, Inc., Provision of Services, Blueprint Medicines, Provision of Services, C4 Therapeutics, Provision of Services, Daiichi Sankyo, Provision of Services, Janssen Pharmaceuticals, Inc., Provision of Services; J.S.R.: Belgian Volition, Provision of Services, Goldman Sachs, Provision of Services, Oncoclinicas do Brasil Servicos Medicos S.A., Fiduciary Role/Position; Ownership / Equity Interests, Paige.AI, Inc., Ownership / Equity Interests; Provision of Services, Personalis, Inc., Provision of Services, Repare Therapeutics, Ownership / Equity Interests; Provision of Services; D.G.: Grail, Provision of Services, Johnson & Johnson, Provision of Services (uncompensated), Med Learning Group, Provision of Services, Medtronic, Provision of Services, Varian Medical Systems, Provision of Services; N.S.: Cambridge Innovation Institute, Provision of Services (uncompensated), Harvard T.H. Chan School of Public Health, Provision of Services (uncompensated), Innovation in Cancer Informatics, Provision of Services (uncompensated), Seoul National University, Provision of Services; L.R.P.: Best Doctors, Provision of Services, Clovis Oncology, Ownership / Equity Interests, Galera Therapeutics, Inc., Provision of Services, Monte Rosa Therapeutics, Inc., Provision of Services, Turnstone Biologics Corp., Provision of Services. The remaining authors declare no competing interests.
