## [Peer Review File · Nature Communications]

REVIEWERS' COMMENTS

Reviewer #1 (Remarks to the Author):

I have reviewed a newer version of the manuscript after I suggested a number of aspects in the initial submission to a different journal of the Nature group. I recognize that my comments were aimed according to the submission to the initial journal. Consequently I faced the current rebuttal with an open mind to judge whether the authors considered whether any of the points initially proposed were adequate to improve the quality of the manuscript. Surprisingly what I found is the recognition of the interest on my comments for the manuscript but a general answer to them stating that they are out of the scope of the current manuscript or that the authors do not perform experiments in mice. Consequently, the only thing I can do as a reviewer is to recognize the value and interest of the manuscript but I cannot assess the revision simply because none of my suggestions have been considered.

Reviewer #2 (Remarks to the Author):

The authors answer to several previous reviewer comments, however almost none of them are incorporated in the manuscript. As outlined by the previous reviewers, the cohort is larger than previous cohorts, however the authors are not able to draw clinically significant further conclusions. Overall, the improvement from approx 70 to low over 100 patients is low. Also the matches pairs with something around 30 is not large. Therefore, there are still several aspects missing like:

- what about patients with unresectable BM - why not include also autopsy specimens - maybe there is a biological bias or mets presenting in a resectable manner
- what about a more in depth analysis of timing and therapy in between samples?
- what on the functional site - what - besides what we already know - is the biological learning - are mutations really that important - isn't it rather the consequence on the functional level to understand why brain mets are growing?
- does the genetic information differ between entities?

Therefore, certainly the data is of interest however not novel, practice changing or informative enough to be published in Nature communications.

Reviewer #3 (Remarks to the Author):

The authors have adequately addressed the referee comments; recognizing the scope of this work focuses on identification of clinical correlates with brain and extra-cranial metastases in matched NSCLC tumors through the MSK-IMPACT study. This will be a valuable resource for the community, which should motivate further basic tumor biology studies to mechanistically dissect some of the observed phenomena. I would only recommend that the paper be formatted better for a journal like Nature Communications, since its current form, is largely aligned with a more clinical journal.

Reviewer #4 (Remarks to the Author):

In this revised manuscript, the authors have added the analysis of the cell cycle pathway and CDKN2A/B alterations to address the reviewers' concerns. However, the reviewer still claims that relationships between the identified mutations, such as CDKN2A/B and HLA-B alterations, and brain metastasis would not be clearly elucidated. The authors should describe (at least discuss) possible molecular mechanisms by focusing on the path to brain metastasis or microenvironment in brain. They can refer to the previous reports/datasets of genomic and transcriptomic analysis related in brain metastasis of lung and other type of cancers.

REVIEWERS' COMMENTS

Reviewer #1 (Remarks to the Author):

I have reviewed a newer version of the manuscript after I suggested a number of aspects in the initial submission to a different journal of the Nature group. I recognize that my comments were aimed according to the submission to the initial journal. Consequently I faced the current rebuttal with an open mind to judge whether the authors considered whether any of the points initially proposed were adequate to improve the quality of the manuscript. Surprisingly what I found is the recognition of the interest on my comments for the manuscript but a general answer to them stating that they are out of the scope of the current manuscript or that the authors do not perform experiments in mice. Consequently, the only thing I can do as a reviewer is to recognize the value and interest of the manuscript but I cannot assess the revision simply because none of my suggestions have been considered.

We want to express our gratitude for your insightful comments during the review process. We appreciate the time and effort you have dedicated to providing such detailed feedback in regards of our work. We assure you that your suggestions were carefully evaluated and thoroughly discussed by our team. While we deeply respect your expertise, we made some difficult decisions during the revision process to ensure the cohesiveness and clarity of this particular work. We realized that with our current study, we will not be able to address them all in a reasonable time.

Reviewer #2 (Remarks to the Author):

The authors answer to several previous reviewer comments, however almost none of them are incorporated in the manuscript. As outlined by the previous reviewers, the cohort is larger than previous cohorts, however the authors are not able to draw clinically significant further conclusions. Overall, the improvement from approx 70 to low over 100 patients is low. Also the matches pairs with something around 30 is not large. Therefore, there are still several aspects missing like:
- what about patients with unresectable BM - why not include also autopsy specimens - maybe there is a biological bias or mets presenting in a resectable manner

We greatly appreciate your comment. Unfortunately, our standard practice does not involve sequencing autopsy samples leading to difficulties in their acquisition. Additionally, proper matched-normal controls like blood are required as MSK-IMPACT utilizes them to eliminate germline variations which would be challenging to obtain postmortem. Finally, the integrity of genomic DNA may be compromised due to variations in sample preparation and storage, as our typical methods may not be employed. This could potentially impact the quality of the stored DNA and introduce biases into the obtained results. In an effort to address the potential confounding effects of tumor size, we compared the genomic profiles of BM specimens that were larger or smaller than 3 cm. As shown in the figure below, there were no statistically significant differences in the frequencies of driver alterations between the two groups.

- what about a more in depth analysis of timing and therapy in between samples?

We thank the reviewer for this suggestion and to address it we investigated the genomic profiles of patients with both a PT/EM sample and a later BM sample, plotting the treatment that was given between sample procurement (see figure below). We saw that the majority of driver alterations (TP53, KRAS, EGFR) were shared between samples. A few patients exhibited private copy-number aberrations in EGFR and CDKN2A in one of their samples, but the majority of alterations that were seen in the PT/EM prior to systemic therapy were seen in the BM sample. Future work will involve prospective evaluation of cohorts on specific treatment such as TKI, incorporating ctDNA into our analysis to track the genomic alterations over time and elucidate whether there are mechanisms of resistance that are arising.

- what on the functional site - what - besides what we already know - is the biological learning - are mutation really that import - isn't it rather the consequence on the functional level to understand why brain mets are growing?

We appreciate your valuable input. So far from previous studies of ours (Choudhury et al., 2022) we know that progression free survival of patients with atypical EGFR mutations is shorter compared to patients with more classical exon 19 deletion or L858R. We also know that their survival is shorter, but it is unclear if this is a predictive or prognostic phenomenon. Although a potential explanation could be reduced drug affinity of Osimertinib for the kinase domain in patients with atypical EGFR mutations, there is currently a lack of solid evidence to substantiate this hypothesis. Furthermore, uncertainty remains regarding the predictive or prognostic significance of atypical EGFR mutations, as it is unclear whether their poor outcomes are directly related to decreased response to targeted therapies or whether

they are prognostic and result in poor outcomes regardless of the treatments given. We believe that future work should include experiments that evaluate structural and allosteric implications of these genomic alterations to evaluate these claims but that is out of scope of this clinical manuscript.

- does the genetic information differ between entities?

While there are a few oncogenic alterations that arise privately between PT/EM and BM tumors, the majority of driver alterations between samples from the same patient were shared across all samples.

Therefore, certainly the data is of interest however not novel, practice changing or informative enough to be published in Nature communications.

Thank you for your thoughtful evaluation of our manuscript. We recognize that Nature Communications sets a high standard for groundbreaking and practice-changing research, and we believe that our study meets these criteria and will be received with interest by the scientific community.

Reviewer #3 (Remarks to the Author):

The authors have adequately addressed the referee comments; recognizing the scope of this work focuses on identification of clinical correlates with brain and extra-cranial metastases in matched NSCLC tumors through the MSK-IMPACT study. This will be a valuable resource for the community, which should motivate further basic tumor biology studies to mechanistically dissect some of the observed phenomena. I would only recommend that the paper be formatted better for a journal like Nature Communications, since its current form, is largely aligned with a more clinical journal.

We thank the reviewer for the time and effort, and we will work with Nature Communications editors on final formatting requests to ensure that we meet all requirements.

Reviewer #4 (Remarks to the Author):

In this revised manuscript, the authors have added the analysis of the cell cycle pathway and CDKN2A/B alterations to address the reviewers' concerns. However, the reviewer still claims that relationships between the identified mutations, such as CDKN2A/B and HLA-B alterations, and brain metastasis would not be clearly elucidated. The authors should describe (at least discuss) possible molecular mechanisms by focusing on the path to brain metastasis or microenvironment in brain. They can refer to the previous reports/datasets of genomic and transcriptomic analysis related in brain metastasis of lung and other type of cancers.

We thank the reviewer for this suggestion and have expanded on potential hypotheses and functional implications of our findings in the discussion, as follows:

"It is an ongoing multi-institutional effort to understand the biological underpinnings of CNS tropism across various cancer types. Common events that appear to be important for metastatic progression include chromosomal instability, impaired DNA repair, copy number alterations and cell cycle alterations. Specifically, copy number deletion of CDKN2A has been one of most frequently reported events¹⁶. CDKN2A can inactivate the RB protein by binding to and inactivating the cyclin D-cyclin-dependent kinase

4 complex. The expression of this gene can cause cell cycle arrest in the G1 phase, inhibit cell proliferation, promote tumor cell apoptosis, and increase tumor cell chemotherapy sensitivity. The current study confirms frequent loss of CDKN2A/B and concordant cell cycle pathway alterations in NSCLC BM. Strikingly, approximately 50% of patients from this cohort had CNA in cell cycle genes that were non-overlapping and mutually exclusive, suggesting that this is a singular event in the development of metastatic disease (Suppl. Fig. 2, D)."